# Mosaic Segmental and Whole-Chromosome Upd(11)mat in Silver-Russell Syndrome

**DOI:** 10.3390/genes12040581

**Published:** 2021-04-16

**Authors:** Laura Pignata, Angela Sparago, Orazio Palumbo, Elena Andreucci, Elisabetta Lapi, Romano Tenconi, Massimo Carella, Andrea Riccio, Flavia Cerrato

**Affiliations:** 1Department of Environmental Biological and Pharmaceutical Sciences and Technologies (DiSTABiF), Università degli Studi della Campania “Luigi Vanvitelli”, 81100 Caserta, Italy; laura.pignata@unicampania.it (L.P.); Angela.sparago@unicampania.it (A.S.); flavia.cerrato@unicampania.it (F.C.); 2Division of Medical Genetics, Fondazione IRCCS “Casa Sollievo della Sofferenza”, 71013 San Giovanni Rotondo, Italy; o.palumbo@operapadrepio.it (O.P.); m.carella@operapadrepio.it (M.C.); 3Medical Genetics Unit, Meyer Children’s Hospital, 50139 Firenze, Italy; elena.andreucci@meyer.it (E.A.); e.lapi@meyer.it (E.L.); 4Department of Pediatrics, Clinical Genetics, Università di Padova, 35122 Padova, Italy; romano.tenconi@unipd.it; 5Institute of Genetics and Biophysics (IGB) “Adriano Buzzati-Traverso”, Consiglio Nazionale delle Ricerche (CNR), 80131 Napoli, Italy

**Keywords:** Silver–Russell syndrome, Beckwith–Wiedemann syndrome, genomic imprinting, imprinting disorders, uniparental disomy

## Abstract

Molecular defects altering the expression of the imprinted genes of the 11p15.5 cluster are responsible for the etiology of two congenital disorders characterized by opposite growth disturbances, Silver–Russell syndrome (SRS), associated with growth restriction, and Beckwith–Wiedemann syndrome (BWS), associated with overgrowth. At the molecular level, SRS and BWS are characterized by defects of opposite sign, including loss (LoM) or gain (GoM) of methylation at the *H19/IGF2*:intergenic differentially methylated region (*H19/IGF2*:IG-DMR), maternal or paternal duplication (dup) of 11p15.5, maternal (mat) or paternal (pat) uniparental disomy (upd), and gain or loss of function mutations of *CDKN1C*. However, while upd(11)pat is found in 20% of BWS cases and in the majority of them it is segmental, upd(11)mat is extremely rare, being reported in only two SRS cases to date, and in both of them is extended to the whole chromosome. Here, we report on two novel cases of mosaic upd(11)mat with SRS phenotype. The upd is mosaic and isodisomic in both cases but covers the entire chromosome in one case and is restricted to 11p14.1-pter in the other case. The segmental upd(11)mat adds further to the list of molecular defects of opposite sign in SRS and BWS, making these two imprinting disorders even more specular than previously described.

## 1. Introduction

SRS (OMIM #180860) is a congenital developmental disorder characterized by heterogeneous clinical features. Currently, the clinical diagnosis of SRS is based on the Netchine–Harbison scoring system, according to which a positive diagnosis is given in the presence of at least four out of the following clinical features: intrauterine growth restriction (IUGR), poor postnatal growth, relative macrocephaly at birth, protruding forehead, body asymmetry, and feeding difficulties. [1].

The genetics of SRS is very heterogeneous and mostly involves imprinted genes [2]. An underlying molecular cause can currently be identified in around 60% of patients with clinical diagnosis of SRS, while in the other 40%, the molecular etiology remains unknown. The most frequent lesions are loss of methylation (LoM) of the: *H19/IGF2*:IG-DMR (also known as IC1) of the telomeric imprinted domain of chromosome 11p15.5 identified in 30–60% of the cases, followed by upd(7)mat detected in 5–10% of the cases. In a subset (∼10%) of the patients with IC1 LoM, hypomethylation involves multiple imprinted loci, a condition also known as multilocus imprinting disturbances (MLID) [3,4]. Concerning the other molecular defects of 11p15.5, maternal duplications of the entire imprinted gene cluster or only the centromeric domain (including the *KCNQ1OT1*:Transcription Start Site-DMR, also known as *KCNQ1OT1*:TSS-DMR or IC2, and the *CDKN1C* gene) are found in less than 1% of the patients, while other 11p15 copy number variants (CNVs), *IGF2* loss of function single nucleotide variants (SNVs), and *CDKN1C* gain of function SNVs are even rarer [5,6,7]. Mosaic upd(11)mat has been reported in only two cases to date, in both of which the whole chromosome was involved [8,9]. Molecular defects affecting further loci are generally found at a low frequency and include genetic mutations of the *HMGA2* and *PLAG1* genes, upd(14)mat, upd(16)mat, upd(20)mat and pathogenic CNVs at several chromosome regions [10,11,12,13,14,15,16].

Here, we report on two patients with mosaic upd(11)mat that is extended to the whole chromosome in one patient, and segmental and restricted to chromosome 11p14.1-pter in the other one.

## 2. Material and Methods

### 2.1. Ethics

Genetic analyses were performed after written informed consent was obtained from the patients or patients’ parents. The research work was carried out in accordance with ethical principles and the Italian legislation. The study was approved by the Ethical Committee of the University of Campania “Luigi Vanvitelli” (Naples, Italy. Approval Number:1135, 13 October 2016).

### 2.2. Genomic DNA Extraction

Genomic DNA was extracted from peripheral blood leukocytes (PBL) using the salting-out procedure, and its concentration was determined using a NanoDrop spectrophotometer (Thermo Fisher Scientific, Waltham, MA, USA).

### 2.3. DNA Methylation Analysis

Methylation-specific multiple ligation-dependent probe amplification (MS-MLPA) was performed on 50 ng of peripheral blood leukocytes (PBL) DNA to analyze DNA methylation and CNVs of several imprinted DMRs by using the SALSA MS-MLPA Kit ME034-B1 (MRC-Holland, Amsterdam, The Netherlands). The amplified products were separated by capillary electrophoresis using an ABI 3500 Genetic Analyzer (Applied Biosystems, Foster City, CA, USA). Data were analyzed using Coffalyser software (MRC-Holland, Amsterdam, The Netherlands).

For the combined bisulfite restriction assay (COBRA), 2 µg of PBL DNA was treated with sodium bisulfite using the EpiTect Bisulfite kit (Qiagen-Italia, Milan, Italy) following the manufacturer’s instructions. Bisulfite-treated DNA was amplified with primers targeting regions within IC1 and IC2. The PCR products were then digested with BstUI (CGCG). Digested (methylated) and undigested (non-methylated) bands were separeted after electrophoresis on a polyacrylamide gel and quantified by phosphorimager to calculate the percentage of methylation. Primer sequences, PCR, and restriction enzyme reaction conditions were previously described [17,18].

Pyrosequencing analysis was carried out as previously reported [19].

### 2.4. SNP Array

Whole-genome CNVs analysis was carried out as previously reported in [20]. The calculation of the mosaic ratio was based on the intensity and distribution of each single nucleotide polymorphism (SNP) allele present on chromosome 11 considering the arbitrary fluorescence unit of haploid locus (single allele) = 0.5 [21]. In the allele difference track, each point represents a SNP interrogated by “A” + “B” allele probes. In physiological conditions, the genotype can be “AA” = [0.5 + 0.5] − [0] = 1, genotype “AB” = [0.5] − [0.5] = 0, genotype “BB” = [0] − [0.5 + 0.5] = −1. Analysis of the allelic differentiation reveals copy neutral mosaicism found in upd as tracts diverging from the common midpoint heterozygote 0 line. Complete chromosome upd is characterized by the absence of a heterozygous state. Mosaic upd is characterized by the presence of lines for AA (intensity = 1) and BB (intensity = −1), the absence of AB, and two new profiles representative of mosaic genotypes AA/AB and AB/BB,. The mix of AB (germline) alleles with converted AA or BB alleles at each SNP generates novel mosaic tracts that represent the relative percentage of upd.

### 2.5. Microsatellite Analysis

Three microsatellites, TH, D11S4088, and D11S922, of chromosome 11p were evaluated in patients and their parents. Primer sequences and PCR condition were obtained from the NCBI Genome Database. PCR amplification of 100 ng DNA was conducted using a forward primer end labeled with Fam or Hex. PCR products were run on the fluorescent capillary system ABI 3500. Data were analyzed using GeneMapper Software (version 5).

## 3. Results

### 3.1. Clinical Cases

The two patients described in this study were referred to our laboratory with a clinical diagnosis of SRS based on the Netchine–Harbison clinical scoring system [1]. The proband of family 1 (II-1 in the pedigree of family 1 in Figure 1) was a 17-year-old boy born after 34 weeks of gestation from healthy unrelated parents. During pregnancy, IUGR and oligohydramnios were reported. Birth weight was 1050 g (<3rd centile), birth length 36 cm (<3rd centile), and head circumference 30 cm (25th centile). At birth, he presented feeding difficulties and respiratory distress. During the postnatal period, feeding difficulties persisted, worsened by the presence of gastro-esophageal reflux. Clinical examination at 1 year revealed skeletal body asymmetry (right part > left part), a small and triangular face with protruding forehead, micrognathia, thin lips, ear abnormalities, fifth finger brachydactyly and clinodactyly, muscular hypotonia, and psychomotor and speech delays. Growth hormone (GH) therapy was undertaken at 6 years because of documented GH deficiency. Chromosome analysis was also performed, revealing a normal 46, XY karyotype. The parents showed no feature of SRS.

The proband of family 2 (II-1 in the pedigree of family 2 in Figure 1) was a 21-year-old man born from non-consanguineous parents. During pregnancy, IUGR was detected. The fetal umbilical blood sampling showed a 46, XY karyotype in 30 metaphases analyzed (G-banding, 300–400 bands). He was born after 35 weeks of gestation, with a birth weight of 1585 g (<3rd centile), length of 38 cm (<3rd centile) and cranial circumference of 31.1 cm (10–25th centile). During the perinatal period, he displayed psychomotor delay and severe feeding difficulties, which persisted during the postnatal period. Hypoglycemia and exceeding sweating occurred in the first year of life. Clinical examination at 6 years revealed skeletal body asymmetry (superior and inferior limbs), a small and triangular face with a protruding forehead, thin lips, ear anomalies, arched palate, cryptorchidism, muscular hypotonia, and motor delay. He is currently attending the second year of university, but he presents motor problems and asthenia. He was treated with GH therapy for about 8 years. No clinical signs of SRS were reported in the parents.

### 3.2. Molecular Diagnosis

Molecular testing for SRS was performed by MS-MLPA on DNA extracted from PBL of the probands, their parents, and unaffected individuals used as controls. In order to test the majority of the molecular defects of SRS through a single approach, we employed the SALSA MS-MLPA Kit ME034-B1 (MRC-Holland, Amsterdam, The Netherlands), which allows one to determine CNVs and DNA methylation of several imprinted loci (*H19*, *KCNQ1OT1*, *PLAGL1*, *GRB10*, *MEST*, *DLK1/MEG3*, *SNRPN*, *PEG3*, and *GNAS/NESPAS*). The analysis of copy number did not reveal any CNVs in either of the two probands (Figure 1). The methylation analysis performed by MS-MLPA revealed IC1 LoM and IC2 GoM at 11p15.5 in the proband 1 (41% and 71%, respectively), while methylation levels comparable to seven unaffected control subjects were detected at these DMRs in the proband 2. Normal methylation levels were found at the other DMRs, thereby excluding MLID in both patients (Figure 1).

To exclude the possibility that methylation abnormalities were undetected in proband 2 because of the relatively low sensitivity of the MS-MLPA, we tested his DNA by using another technique, the COBRA. By employing this method, a slight IC1 LoM and a slight IC2 GoM were found in the proband 2 DNA compared to that of his parents and two controls (Appendix A). No CNVs and normal methylation levels of the analyzed regions were found in the parents of both probands (Appendix A). The slight methylation defect of the two 11p15.5 DMRs was further confirmed in both the probands by using a third approach, the bisulfite treatment of DNA followed by pyrosequencing (Figure 2 and Appendix A), which was performed to analyze, in addition to IC1 and IC2, *MEST*:alt-TSS-DMR and *GRB10*:alt-TSS-DMR as controls.

In summary, the opposite methylation defects of the 11p15.5 DMRs without evidence of CNVs suggest the presence of maternal upd in both proband 1 and proband 2.

### 3.3. Characterization of the Molecular Defects

To better characterize the molecular defects of the probands, we further analyzed their PBL DNA by employing the SNP array. Mosaic upd(11)mat was identified in the probands 1 and 2 (Figure 3). Interestingly, in proband 1, upd was segmental and restricted to chromosome region 11p14.1-pter, while in proband 2 the upd was extended to the whole chromosome 11. The percentage of mosaicism was 20% in the proband 1 and ∼10–15% in the proband 2. No upd pathogenic CNVs were identified in proband 1’s parents by SNP array analysis (Appendix A). The parents of proband 2 could not be analyzed because the required DNA was unavailable. The analysis of three informative short tandem repeats (STR) markers (D11S922, TH, D11S4088) indicated the presence of maternal uniparental isodisomy (UPiD) of chromosome 11p15.5 in both probands 1 and 2 and excluded the heterodisomy (Table 1).

## 4. Discussion

Lesions at the 11p15 imprinted gene cluster represent the majority of molecular defects associated with SRS to date. They include the frequent IC1 hypomethylation, the rare CNVs, and the extremely rare upd(11)mat and *IGF2* and *CDKN1C* mutations. We have identified two novel cases with mosaic maternal UPiD 11 and the typical SRS phenotype. In one of the patients, the maternal UPiD 11 is restricted to 11p14.1-pter and represents the first reported case of such molecular defect in SRS.

The segmental upd(11) mat adds further to the list of molecular defects of opposite sign in SRS and BWS, making these two imprinting disorders even more specular than previously described. However, some differences are worth mentioning. Mosaic upd(11)pat is a frequent molecular defect in BWS affecting 20% of the cases [22], most of which (78–92%) are reported to have a segmental upd 11p [23,24]. Conversely, mosaic upd(11)mat is extremely rare in SRS, and of the four cases reported to date, three of them have whole-chromosome upd(11)mat.

Upd can arise as meiotic or mitotic errors [25,26]. In our probands, somatic mosaicism and isodisomy suggest post-fertilization errors that might derive from mitotic nondisjunction followed by trisomic rescue in the case of complete upd or mitotic recombination between non-sister chromatids in the case of segmental upd. The latter is assumed to be the most probable event causing mosaic segmental paternal UPiD 11 in BWS. It should result in the formation of two daughter cells, one with maternal isodisomy and the other one with paternal isodisomy. The cells with upd(11)pat are expected to have a selective advantage over the upd(11)mat because of the altered dosage of imprinted genes controlling cell growth (including *IGF2*, *H19*, and *CDKN1C*) [26,27]. The finding of segmental upd(11)mat may be explained by the presence of a recessive mutation on the paternal chromosome, resulting in the negative selection of the cells with paternal isodisomy. Alternatively, other somatic recombination events similar to gene conversion might be responsible for the formation of segmental maternal UPiD 11 [28].

The mosaic whole-chromosome UPiD 11 of both SRS and BWS is likely derived from mitotic nondisjunction followed by trisomic rescue. Indeed, loss of one of the supernumerary chromosomes is a prerequisite for the survival of trisomic embryos. In one-third of these cases, the homolog that is present in single copy is lost, thus giving rise to a upd [25]. It is plausible that the chromosome that is lost is the maternal or paternal one by chance, determining mosaic complete maternal or paternal UPiD 11 with the same probability. Consistent with this hypothesis, this defect is similarly rare in SRS and BWS [8,9,23,24].

Upd 11 has been reported in a mosaic form only to date, suggesting that the non-mosaic form is incompatible with life, because an altered dosage of the 11p15 imprinted genes is tolerated only if restricted to a subset of somatic cells [26,29]. Similar to the previously reported cases of upd(11)mat, the percentage of mosaicism in our patients is very low (10–20%) [8,9] and generally lower than that reported in BWS cases [23,24]. This suggests that differently from upd(11)pat, upd(11)mat is compatible with life only if it affects a very restricted number of cells. For this reason, it may be that mosaic upd(11)mat is a more common cause of SRS than was currently observed, since it is difficult to detect, either for technical reasons or because it involves tissues other than the leukocytes sampled for testing [9].

At the clinical level, apart from the micrognathia and down-turned mouth, no main difference can be found between the case with segmental and those with whole-chromosome upd(11)mat (present case and those reported in [8,9]) (Table 2). No major phenotypic difference has been reported also between segmental and whole-chromosome upd(11)pat in BWS; thus, it has been proposed that, rather than the extension of the upd, the proportion of the cells and the kind of tissues affected is responsible for the variability of the clinical manifestations [24,30,31]. Concerning the SRS molecular subgroups, the main phenotypic difference is body asymmetry (Table 2). Body asymmetry is present in the patients described in the present study and the other reported mosaic upd(11)mat as well as in 77% IC1 LoM cases, but only in 29% upd(7)mat cases, and it is generally absent in 11p15 duplication cases. These differences are likely due to mosaicism that is always associated with upd 11, frequently associated with IC1 LoM, and usually absent in upd(7)mat and 11p15 duplication cases. No further clinical feature appears to be significantly different in its frequency between the upd(11)mat and the other SRS molecular subgroups.

## 5. Conclusions 

In conclusion, we reported on two novel SRS cases with the extremely rare mosaic upd(11)mat. In one of these cases, upd(11)mat is extended to the whole chromosome and in the other is restricted to 11p14.1-pter. Such finding increases the list of molecular defects of opposite sign in SRS and BWS, although in BWS upd(11)pat is relatively common and generally segmental, while in SRS upd(11)mat is rare and more frequently affecting the entire chromosome. The rare case of mosaic segmental upd(11)mat in SRS is likely caused by mitotic recombination associated with the presence of a recessive mutation on the paternal chromosome resulting in the negative selection of the upd(11)pat cells.

## Figures and Tables

**Figure 1 genes-12-00581-f001:**
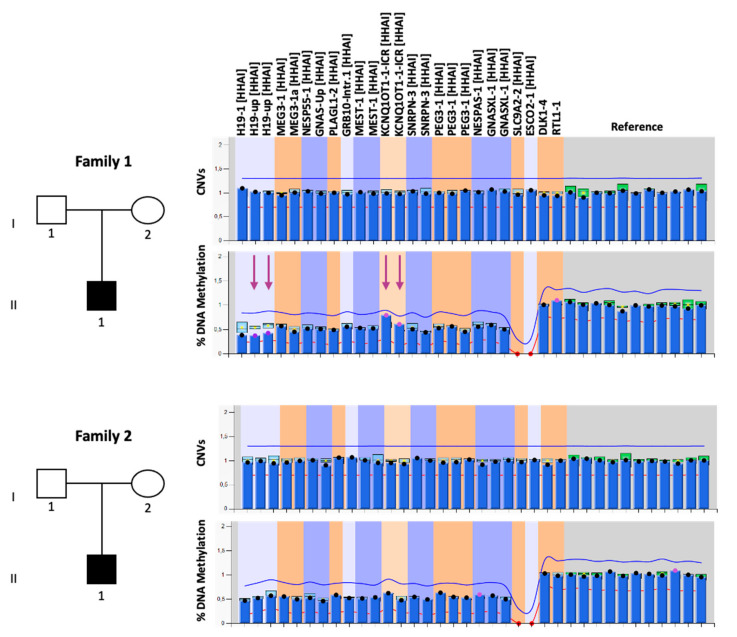
Genetic characterization of the probands and their families. Left: pedigrees of the two families. Right: MS-MLPA results. CNVs (upper panels) and DNA methylation (lower panels) were analyzed at 10 imprinted loci, reported at the top of the figure. The mean values of seven control subjects were utilized for the assessment of relative copy number and methylation percentage. The purple arrows indicate the affected regions.

**Figure 2 genes-12-00581-f002:**
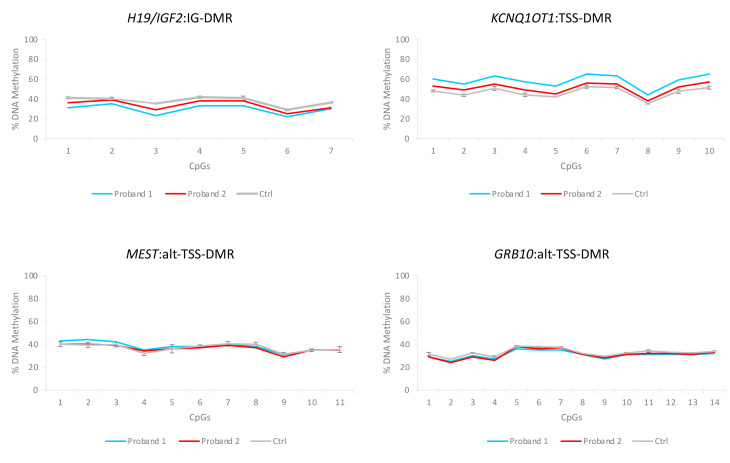
DNA methylation analysis of four DMRs performed on proband 1 and proband 2. Ctrl: average of three unaffected individuals.

**Figure 3 genes-12-00581-f003:**
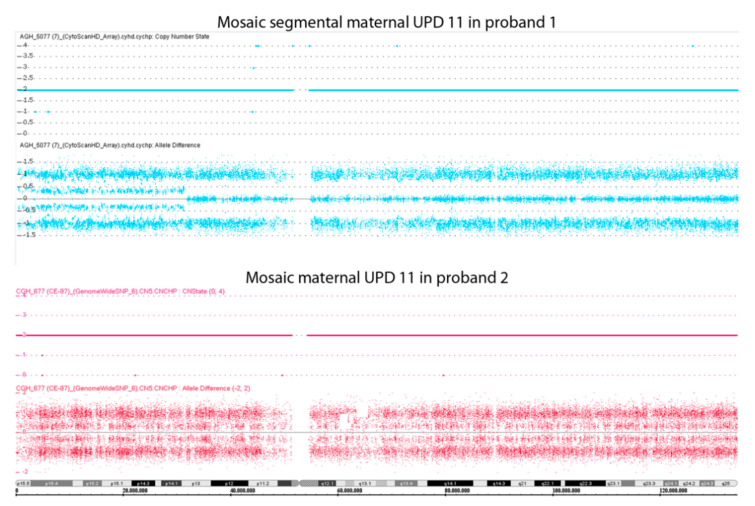
SNP array analysis on genomic DNA of the two probands. Each panel represents the SNP array results of chromosome 11 only. The upper graphs indicate the copy number state; the lower graphs indicate the B allele frequency for each SNP. Note the differences presented by these two cases on the extent of upd, partial in proband 1 and complete in proband 2, and their level of mosaicism, which is higher for proband 1 (20%) than for proband 2 (∼10–15%).

**Table 1 genes-12-00581-t001:** Summary of genotypes of probands with mosaic upd 11 and their parents for three short tandem repeats (STRs) on chromosome 11p15.5.

STR	Family 1	Family 2
D11S922	bc*, ac, bd	aa, ab, ab
TH	a*c, ab, cd	b*c, ab, cc
D11S4088	a*c, ab, cd	c*d, ac, bd

The alleles are given in the following order: patient, mother, father. Allele designations (marked a to d) are arbitrary. The asterisk indicates the allele with higher intensity in the probands.

**Table 2 genes-12-00581-t002:** Frequency of clinical features in the main SRS molecular subgroups.

Clinical Sign	Proband 1 (Segmental)	Proband 2 (Whole-Chromosome)	Proband Described in [8] (Whole-Chromosome)	Proband Described in [9] (Whole-Chromosome)	11p15 LoM (%) *	Upd(7)mat (%) *	11p15 dup (%)
Sex	Male	Male	Female	Male			
^1^ SGA: birth weight and/or birth length	Yes	Yes	Yes	Yes	100	73	100 **
Postnatal growth failure	Yes	Yes	Yes	Yes	84	81	100 **
Relative macrocephaly at birth	Yes	Yes	No	Yes	99	85	95 **
Protruding forehead	Yes	Yes	Yes	Yes	94	100	90 **
Body asymmetry	Yes	Yes	Yes	Yes	77	29	0 **
Feeding difficulties and/or low BMI	Yes	Yes	Yes	Yes	72	87	88 **
Triangular face	Yes	Yes	Yes	Yes	99	50	74 ***
Fifth finger clinodactyly	Yes	No	Yes	Yes	81	56	93 ***
Micrognathia	Yes	No	No	No	75	26	nr ***
Low muscle mass	Yes	Yes	No	No	67	47	67 ***
Excessive sweating	No	Yes	Yes	No	51	70	nr ***
Down-turned mouth	Yes	No	No	No	57	26	nr ***
Genital abnormalities	No	Yes	No	Yes	nr	nr	nr ***
Speech delay	Yes	No	No	Yes	32	64	78 ***
Motor delay	Yes	Yes	Yes	No	30	58	

^1^ SGA: small for gestational age; BMI: body mass index; the extension of the upd in each proband is reported in brackets. Cases reported in * [2], ** [32,33,34], *** [33,34].

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
