# Peer review of "Mosaic Segmental and Whole-Chromosome Upd(11)mat in Silver-Russell Syndrome"

_genes, 2021, doi:10.3390/genes12040581_

Round 1
Reviewer 1 Report
This is well written and well documented description of two important cases that clarify and extend our understanding of the molecular changes that lead to SRS.
I have very little to add, except for 4 minor corrections and 2 requests for clarification:
L31 "...more specular than what previously described." Delete "what"
L72 "peripheral blood lymphocytes (PBL)". It is unusual to purify lymphocytes for the purpose of DNA extraction. Should this is "leukocytes" or "mononuclear cells" (if prepared by density layering).
L87 BstUI. Refer to RJ Roberts NAR 2003, regarding RE nomenclature - no italics please.
L114 State whether GH deficiency documented or assumed clinically or unknown?
L218 spelling tolerated. Should be tolerated
L229 spellng extention. Should be extension
Reviewer 2 Report
The authors reported the SRS cases with maternal uniparental disomy of chromosome 11. The reviewer thinks this paper is clinically important, however, the reviewer has some comments.
Comment 1
Although proband 1 had mosaic segmental upd(11)mat, proband 2 had mosaic full upd(11)mat. The reviewer thinks that the title of this manuscript is not suitable.
Comment 2
Regarding the results of methylation analysis in proband 2 shown in Figure 2, it was difficult to detect hypomethylation of the IGF2/H19:IG-DMR and hypermethylation of the KCNQ1OT1:TSS-DMR. The authors need to perform another methylation analysis to show the abnormal methylation status of both DMRs in proband 2. Bio-COBRA which use bioanalyzer to quantile the methylation levels of the DMRs and bisulfite sequencing have been reported [2008, Yamazawa k et al. J Mol Med (Berl)] .
Comment 3
For Figure 3, the authors need to use high resolution images.
Comment 4
For Table 1, to estimate the mosaic ratio, the reviewer think that the raw data of the microsatellite analysis which showed small peaks inherited from the father in some markers are informative.
Comment 5
In Table 2, the addition of the previously reported the frequencies of the clinical features in the cases with LoM of the IGF2/H19:IG-DMR, UPD(7)mat, and 11p15.5 duplication will support to understand of the clinical features in the cases with mosaic UPD(11)mat. In addition, to clarify the meaning of the genital abnormalities, the authors need to add the information of the sex of cases with mosaic UPD(11)mat. The reviewer thinks that the clinical features of the cases with mosaic UPD(11)mat showed typical SRS features. The authors need to discuss about the SRS clinical features in mosaic UPD(11)mat in more detail.
Comment 6
The authors need to describe the way of calculation of the mosaic ratio from SNP array data.
Comment 7
The reviewer wants to know the results of karyotyping of these cases. Did authors perform karyotyping in these cases?
Round 2
Reviewer 2 Report
The authors revised their manuscript based on my comments. I think that this paper should be accepted.